# Thermoregulatory Dysfunction in Parkinson’s Disease: Mechanisms, Implications, and Therapeutic Perspectives

**DOI:** 10.3390/cells14231910

**Published:** 2025-12-02

**Authors:** Zechariah S. Pressnell, Lee E. Neilson, Domenico Tupone, Ronald F. Pfeiffer, Delaram Safarpour

**Affiliations:** 1Undergraduate Department of Biology, Washington University in St. Louis, St. Louis, MO 63130, USA; p.zechariah@wustl.edu; 2Neurology and Research Service, VA Portland Health Care System, Portland, OR 97239, USA; neilsole@ohsu.edu; 3Department of Neurology, Oregon Health & Science University, OP-32 3181 SW Sam Jackson Park Rd., Portland, OR 97239, USA; pfeiffro@ohsu.edu; 4Department of Neurology, University of Iowa, Iowa City, IA 52242, USA; 5Department of Neurosurgery, Oregon Health & Science University, Portland, OR 97239, USA; 6Department of Biomedical and Neuromotor Science, University of Bologna, 40126 Bologna, Italy

**Keywords:** Parkinson’s disease, thermoregulation, autonomic dysfunction, hyperhidrosis, hypohidrosis, heat intolerance, α-synuclein, Parkinsonism–hyperpyrexia syndrome

## Abstract

**Highlights:**

**What are the main findings?**
Thermoregulatory dysfunction in Parkinson’s disease (PD) arises from combined central and peripheral α-synuclein pathology, disrupting hypothalamic integration, sudomotor, vasomotor, and thermogenic pathways.Distinct alterations in sweating, heat and cold tolerance, and crisis syndromes such as parkinsonism-hyperpyrexia and hypothermia underscore the multisystem nature of PD’s autonomic impairment.

**What are the implications of the main findings?**
Recognizing and managing thermoregulatory dysfunction can improve patient safety, reduce hospitalization risk, and enhance quality of life in PD.Future studies should investigate how PD affects brown adipose tissue (BAT) and shivering thermogenesis, offering potential targets for restoring heat balance and metabolic stability.

**Abstract:**

Thermoregulatory dysfunction—temperature intolerance and/or inappropriate compensation—is an underrecognized feature of Parkinson’s disease (PD) and is linked to poor quality of life. Multiple mechanisms may underlie this dysfunction, including α-synuclein deposition in relevant structures, altered functional connectivity in thermoregulatory networks, and disrupted neurotransmitter modulation, on top of the deleterious consequences of aging. Although multiple advanced tests can confirm this dysfunction, diagnosis is largely based on a detailed history. Once this critical symptom is identified, management focuses on crisis prevention and safety, as PD-specific clinical trials are often lacking. This narrative review of the literature addresses mechanisms, clinical expression, diagnostic evaluation, and management of thermoregulatory dysfunction in PD to help guide care for this underappreciated, yet potentially debilitating, non-motor symptom of PD. Future PD-specific trials are needed to further clarify underlying mechanisms and improve treatment options.

## 1. Introduction

James Parkinson, in his 1817 characterization of what we now know as Parkinson’s disease (PD), noted the presence of autonomic disturbances in the form of gastrointestinal and urinary dysfunction [1]. Subsequently, in 1898, William Gowers described thermoregulatory dysfunction as the sensation of coldness experienced by some patients in their affected limbs [2]. Additional autonomic abnormalities, including cardiovascular and sexual dysfunction, also were identified in PD. These early writings foreshadowed modern recognition of PD as a multisystem disorder involving both motor and non-motor symptoms, including sleep, behavioral, sensory and autonomic dysfunction [3,4], with thermoregulatory abnormalities now recognized as far more clinically significant than when first alluded to by Gowers.

Normal thermal homeostasis depends on peripheral thermal sensors, central neuronal integrators, and descending efferents coordinating thermo-effectors for sweating, vasodilation, muscle shivering thermogenesis and brown adipose tissue (BAT) [5,6,7,8,9].

Thermoregulatory dysfunction in PD manifests with a broad spectrum of clinical dysfunction: hyperhidrosis, hypohidrosis, heat and cold intolerance, and, in severe cases, crises characterized by hyperpyrexia and hypothermia [10,11]. Recent literature stresses that these disturbances extend beyond quality-of-life impairment to include increased vulnerability to environmental extremes, perioperative instability, morbidity, and hospitalization risk [12,13]. These concerns are magnified by climate change projections of more frequent heatwaves [14].

While aging itself diminishes thermoregulatory function, thermoregulation and autonomic function may be further compromised in PD by deposition of aggregated misfolded α-synuclein proteins that form Lewy bodies and Lewy neurites—the pathological hallmarks of PD—in specific autonomic neuronal centers. α-synuclein deposition affects both sympathetic and parasympathetic nuclei, including hypothalamic preoptic regions, brainstem autonomic centers, intermediolateral spinal columns, and postganglionic efferents, and may contribute to the functional alteration of these centers in PD. Indeed, some functional and morphologic studies confirm postganglionic sudomotor deficits in PD, with α-synuclein deposition in postganglionic cutaneous fibers innervating sweat glands [15,16].

Related disorders also marked by α-synuclein deposition, such as multiple system atrophy [MSA), present with autonomic dysfunction [17,18]. However, although MSA shares some clinical features of thermoregulatory dysfunction with PD, its predominantly preganglionic pathology [17,18] and distinct α-synuclein distribution make it a separate entity. Given the limited research specifically addressing thermoregulation in MSA, this review will focus primarily on PD/Dementia with Lewy Bodies (DLB) while noting that overlapping mechanisms among these pathologies remain plausible.

Nevertheless, α-synuclein structural vulnerabilities provide a mechanistic basis for the thermoregulatory dysfunction characteristic of autonomic involvement in PD and related synucleinopathies [10,11,12,13,14].

## 2. Normal Thermoregulatory Function

Thermoregulation is an important homeostatically controlled function necessary to maintain the optimal core body temperature, which in humans is roughly 37.5 °C. At this temperature, all physiologically regulated functions in the human body operate at their most efficient state. The thermoregulatory systems guarantee normal thermoregulation by a coordinated neuronal structure formed by three interacting components: (1) thermal sensing by peripheral cutaneous and visceral afferents; (2) hypothalamic central integration of peripheral thermal information with central thermoreceptors and other physiological regulated functions (e.g., osmolarity, metabolism) to defend body thermal homeostasis by recruiting downstream pathways for the activation/inhibition of the (3) autonomic thermoeffectors, such us skin vasculature and sweat glands, to conserve/dissipate heat; BAT for production of heat (thermogenesis); and somatic thermoeffector skeletal muscle for shivering thermogenesis.

Higher cortical networks shape thermoregulatory behavior, such as seeking shade or adjusting clothing, providing a voluntary route that complements autonomic outputs. The sections that follow first outline thermosensitivity, then describe the effector mechanisms, and finally summarize the integrating network that links them, setting the stage for disease-related alterations.

### 2.1. Thermosensitivity

Thermosensitivity is the inherent ability of certain neurons to detect changes in temperature by increasing their neuronal activity in a linear correlation with thermal variation. There are two main types of thermosensitive neurons: those responding to cold and those responding to warm thermal stimuli. Their thermal sensitivity is primarily due to the presence of specific receptors known as transient receptor potential (TRP) channels. Several kinds of TRP channels exist; some respond to mild cold or warm stimuli and play a key role in thermoregulation, while others respond to extreme temperature changes and are responsible for signaling noxious (painful) thermal stimuli. TRP channels, such as TRPV1 for heat and TRPM8 for cold, play central roles in encoding thermal stimuli [19,20,21]. Recent molecular studies have added complexity to this view: TREK-1 and TREK-2, members of the K2P potassium channel family, were shown to act as inhibitory thermosensors. Their gating relies on phosphorylation states and interactions with the microtubular cytoskeleton, with protein kinase A (PKA) phosphorylation serving as a molecular switch that suppresses thermosensitivity [22]. This indicates that inhibitory signaling, alongside excitatory TRP-dependent activity, shapes thermoregulatory responses. In the following discussion, we will focus on cold-responsive channels, which mediate thermoregulatory responses.

Thermosensitive neurons can also be classified into two categories:(1)Peripheral thermosensitive neurons, located in the somatic and visceral layers of the body, which provide thermal information to higher brain structures via second-order neurons.(2)Central thermosensitive neurons, located within central structures of the brain, which monitor and provide local thermal information.

Peripheral thermosensation is a major contributor to the autonomic and behavioral responses to changes in ambient temperature and represents an important anticipatory mechanism that prevents the core body temperature from deviating significantly from its optimal level.

From a peripheral perspective, thermal sensing begins in cutaneous primary afferents. Recordings from dorsal root and trigeminal ganglia identify cold-responsive small-diameter C fibers and medium-diameter Aδ fibers that either fire tonically near normal skin temperatures and increase discharge with cooling (cool thermoreceptors) [19]. In vitro, roughly 10–15% of sensory neurons respond to cooling with activation thresholds spanning ~35 °C down to ~15 °C, reflecting the wide behavioral range of innocuous–to-noxious cold [19,21]. Molecular work implicates cold-transduction ion channels in these afferents, endowing subsets of neurons with the ability to generate signals at low temperature [19]. Skin and craniofacial inputs terminate on modality-specific lamina I neurons (COOL and WARM classes) that ascend in labeled-line fashion via the lateral spinothalamic tract to a posterolateral thalamic relay and onward to a somatotopic map of temperature in dorsal posterior insula, linking thermosensation to affect, motivation, and thermoregulatory behavior [5]. Together, these skin endings provide finely graded information about ambient temperature changes and trigger thermoregulatory adjustments.

Central thermo-sensitivity relies on the presence within the preoptic area (POA) of the hypothalamus of specific neuronal populations capable of increasing their firing rate as temperature rises, when studied in an in vitro preparation with synaptic blockade to eliminate the potential influence of thermally responsive afferents. Thermo-sensitive neurons are distinguished by thermo-responsive neurons which are activated by increases in ambient environmental temperature or inhibited by skin cooling solely by virtue of their synaptic connections in thermoregulatory networks.

Within the POA, an intricate thermoregulatory neuronal network, that includes a subset of warm-sensitive neurons (WSN) (Figure 1), is considered the central integrator necessary for the homeostatic control of body temperature. WSN are activated by increases in ambient temperature by virtue of their synaptic connections within thermoregulatory networks (thermo-responsive) but they also respond to local increases in brain temperature due to their intrinsic thermosensitivity [23]. Conceptually, these POA neurons operate within a broader thermoregulatory architecture in which skin (shell) inputs provide rapid auxiliary feedback, while deep (core) temperature supplies the principal negative feedback that drives effector recruitment [7]. The thermoregulatory network is also very plastic. For example, prolonged heat exposure remodels these circuits: adult hypothalamic neurogenesis arising from third-ventricle progenitors is induced during long-term heat acclimation [24,25].

### 2.2. Thermoeffectors

To maintain thermal homeostasis, autonomic and somatic regulated thermal effectors act in concert to conserve or dissipate heat, through mechanisms such as vasomotion, sweating, panting, and piloerection, or to generate heat via thermogenesis. The cutaneous vasculature serves as the principal pathway for dry heat exchange during both cold and warm exposure. Sympathetic noradrenergic outflow induces vasoconstriction during cold exposure, reducing skin blood flow and thereby limiting convective and radiative heat loss [26]. Conversely, during heat exposure, sympathetic vasoconstrictor activity is suppressed, promoting vasodilation and facilitating heat loss. An active vasodilator system that relies on cholinergic transmission augmented by nitric oxide, vasoactive intestinal peptide, and prostaglandins to markedly increase skin perfusion has also been described [27]. These perfusion changes are most pronounced in the distal extremities, where regulation of blood flow strongly influences overall heat conservation and dissipation [26,27].

Under more severe cold conditions, shivering of skeletal muscles becomes the dominant thermogenic response. Descending commands from hypothalamic and medullary centers activate somatic motor neurons rhythmically (producing shivering), which rapidly increases metabolic muscle heat production (shivering thermogenesis) [28,29]. Complementing this, β3-adrenergic receptor–dependent activation of BAT, a specialized thermogenic organ, mediates non-shivering thermogenesis through a mitochondrial proton leak mechanism driven by uncoupling protein-1 (UCP1), which dissipates the proton gradient normally used for ATP synthesis and thereby generates heat [30]. BAT thermogenesis is directly regulated by the sympathetic nervous system, whereby norepinephrine release not only controls the magnitude of heat production during acute, transient cold exposure but also mediates adaptation to sustained cold by increasing UCP1 expression and promoting the ‘browning’ of white adipose tissue. BAT thermogenesis may play a critical physiological role in regulating temperature, especially when shivering systems are impaired by pathological conditions (e.g., muscle atrophy) [30].

On the other end, during heat exposure, shivering and BAT thermogenesis are inhibited, which prevents heat production and facilitates reduction in body temperature.

Finally, in humans, eccrine sweat glands provide the most widespread means of evaporative heat dissipation, operating across nearly the entire body surface. Preganglionic neurons in the intermediolateral spinal cord project to postganglionic sympathetic cholinergic fibers that release acetylcholine onto muscarinic M3 receptors of eccrine secretory cells, initiating sweat secretion [6,31]. Together, these thermoeffectors provide a flexible and redundant system, that enables both rapid, short-term adjustments and prolonged defense of body temperature under environmental and metabolic demanding challenges (e.g., exercise).

It is important to note that the thermoregulatory system and the thermoeffectors are strongly influenced by aging and pathological states (e.g., fever, PD) that can compromise thermoregulatory functions, such as sweating and shivering [32,33]. For instance, BAT is highly active in neonates and persists into adulthood, but its activity declines with age and obesity [30,32].

### 2.3. Thermoregulatory Network

The main thermoregulatory network responds to changes in the discharge of thermoreceptors in the skin, muscle, and viscera to regulate body temperature. Peripheral sensory input is conveyed by dorsal root and trigeminal ganglion afferents, to the dorsal horn of the spinal cord, and to the parabrachial nucleus (PBN) in the brainstem. Thermal information from the PBN is then conveyed to the POA, providing control of the descending thermoregulatory networks for the activation or inhibition of thermoeffectors [8,9,34]. Parallel to the main PBN to POA pathway, recent discoveries have demonstrated that direct inputs to the DMH from thermoresponsive PBN neurons [35,36] also play a role in normal thermoregulation [35,36]. The POA consequently intersects with many other hypothalamic regions, including the medial preoptic area (MPA), the median preoptic area (MnPO), the ventromedial periventricular area (VMPeA), the lateral hypothalamus (LH), the dorsal hypothalamic area (DA), and the dorsomedial hypothalamus (DMH) [34,35,37,38,39,40,41,42,43,44,45]. Direct or indirect descending hypothalamic efferents influence the discharge of neurons in the rostral raphe pallidus (rRPa), which in turn provide input to the intermedial lateral columns and the sympathetic ganglia for the control of vasomotion and BAT thermogenesis, as well as input to the ventral horn motor neurons for control of muscle shivering thermogenesis [45,46].

Of particular interest is the newly described concept of thermoregulatory inversion (TI) [35,47], in which the normal thermoregulatory responses are reversed—such that cold exposure inhibits thermogenesis, while warm exposure activates it. TI is normally initiated by inhibition of the POA, and the inverted, skin-evoked thermoregulatory responses are mediated through an alternative pathway that from PBN directly control the thermogenic-driving neurons in the DMH [35]. TI represents an intriguing and paradoxical thermoregulatory paradigm that may help explain some of the altered thermoregulatory responses observed under specific pathological conditions [47].

This complex thermoregulatory system also integrates with other homeostatic controllers. Osmolarity regulation overlaps with thermoregulation through hypothalamic circuits adjacent to the POA, where vasopressin- and angiotensin-responsive neurons prioritize water balance, often attenuating sweating and vasodilation during dehydration [6,7]. Similarly, metabolic inputs, including glucose availability and hormones such as leptin and thyroid hormones, modulate thresholds for BAT thermogenesis and shivering, linking energy balance with temperature defense [7,23]. These interactions emphasize that thermoregulation is not an isolated system but a tightly meshed component of the broader homeostatic network [8].

Taken together, this framework predicts, and clinical data confirm, that in PD, thermoregulatory function can be disrupted at all three levels. At the thermal-sensation level, peripheral small-fiber and sudomotor involvement with cutaneous α-synuclein pathology alters afferent signaling from the skin and contributes to abnormal temperature perception and sweating patterns summarized in PD cohorts [48,49]. At the level of the thermoregulatory network, Lewy pathology involving hypothalamic and brainstem autonomic centers impairs POA–DMH/raphe integration and descending control of effectors [50,51]. Finally, thermoeffectors themselves are frequently abnormal, resulting most prominently in sudomotor dysfunction with distal, length-dependent hypohidrosis or mixed patterns, and vasomotor dysregulation, producing heat/cold intolerance and, in some patients, crises. These patterns also help distinguish PD’s predominantly postganglionic sudomotor deficit from the more preganglionic profile typical of MSA [17,18,48,49,52]. In short, PD can affect thermal sensation, central network integration, and thermoeffector execution, aligning with the mechanisms and pathways outlined above [12,51].

### 2.4. Circadian, Sleep, and Aging Influences

Thermoregulatory outputs are modulated by temporal and lifespan-dependent factors. Circadian rhythms govern daily body temperature cycles, with core temperature declining at sleep onset and rising toward wakefulness [53]. Distal skin vasodilation promotes heat loss and sleep initiation, while ultradian rhythms produce smaller fluctuations across vigilance states [54]. Sleep stages are tightly linked to body and skin temperature oscillations [55]. Aging introduces diminished thermoeffector capacity: reduced sweating, impaired vasodilation, and decreased BAT activity collectively blunt the ability to maintain homeostasis during thermal extremes [32]. These changes increase susceptibility to heat- and cold-related illnesses in older populations.

### 2.5. Behavioral and Higher-Order Regulation

Thermoregulation extends beyond autonomic effectors into motivational and affective domains. Cortical areas, including the insula, anterior cingulate, and orbitofrontal cortex integrate interoceptive signals with emotional states, producing sensations of comfort or discomfort [5,29]. These sensations drive behaviors such as adjusting clothing, altering activity levels, or seeking shelter, which form essential components of thermoregulation alongside autonomic responses [7,54].

## 3. Thermoregulatory Pathophysiology in Parkinson’s Disease

The major mechanisms of thermoregulatory pathophysiology include α-synuclein deposition in relevant structures, altered functional connectivity in thermoregulatory networks, and disrupted neurotransmitter modulation [10,49]. Mapping mechanism to phenotype remains incomplete. Most links between α-synuclein deposition, alterations in thermoregulatory network connectivity, and dopaminergic tone with clinical thermoregulatory signs are primarily associative, drawn from cutaneous and central pathology studies, physiologic testing, and experimental models. This means effect sizes likely vary by disease stage, comorbidities, and medication status.

The pathophysiology of hypohidrosis in PD reflects both peripheral and central mechanisms. This is accompanied by loss of intraepidermal and sudomotor nerve fiber density, changes that correlate with objective measures of reduced sweating. Central contributions likely include degeneration within hypothalamic preoptic and brainstem autonomic nuclei, as well as altered cortical–autonomic network connectivity [17]. The combined effect is impaired initiation and modulation of sweating, with peripheral denervation compounding central command deficits. Peripheral contributions include α-synuclein deposition in cutaneous autonomic fibers innervating sweat glands, likewise demonstrated in skin biopsy studies, supporting a primary postganglionic lesion [15,16,48]. The frequent association of truncal hyperhidrosis with extremity hypohidrosis suggests a compensatory redistribution of sweating [56].

Beyond sudomotor failure, thermodysregulation in PD reflects multisystem synucleinopathy involving central autonomic nuclei and peripheral autonomic pathways, providing substrates for both heat-loss (sweating/vasodilation) and heat-conservation (vasoconstriction) defects [57]. Clinical series and reviews highlight heterogeneous patterns characterized by distal hypohidrosis with truncal hyperhidrosis, and heat and cold intolerance, consistent with combined sudomotor and cutaneous vasomotor involvement [49]. Physiologic studies demonstrate reduced sympathetic skin responses and impaired skin blood-flow regulation, indicating abnormalities in both cholinergic sudomotor and adrenergic vasomotor arms [58]. Observations from related synucleinopathies underscore autonomic pathway involvement that can reduce acral perfusion (“cold limb” phenomena) while coexisting with sweating abnormalities, supporting a shared mechanism of brainstem and peripheral autonomic dysfunction [59].

Furthermore, experimental models demonstrate that dopamine interacts with other neurotransmitters in the preoptic anterior hypothalamus to regulate thermogenesis, with responses dependent on the prior thermal state. Thus, dopamine can induce either hypothermia or hyperthermia depending on baseline conditions [50,60].

## 4. Manifestations of Thermoregulatory Dysfunction in Parkinson’s Disease

### 4.1. Hyperhidrosis

Hyperhidrosis, excessive sweating beyond thermoregulatory needs, is one of the most common and clinically significant manifestations of thermoregulatory dysfunction in PD, with prevalence estimates ranging from 5.5 to 12.9% in newly diagnosed, untreated patients to over 60% in those in advanced disease stages [49]. The highest prevalence is evident in patients experiencing motor fluctuations such as dyskinesia, or wearing-off phenomena [10,48]. Longitudinal data confirm rapid early escalation, with some reports indicating prevalence doubling from 12.9% at diagnosis to 25.8% within one year, and others noting an increase from 28.2% to 35.3% over two years—underscoring progression alongside motor complication development [49,61,62]. Hyperhidrosis may exert a substantial quality-of-life burden. Patients report heat intolerance, social embarrassment, clothing changes, sleep disruption, and activity avoidance [48,63]. Hyperhidrosis often clusters with other dysautonomic features, including orthostatic hypotension, urinary urgency, constipation, sialorrhea, and disproportionate fatigue [10,49]. Disease subtype may influence the risk of developing hyperhidrosis, since hyperhidrosis is more prevalent in patients with high overall non-motor burden, possibly reflecting a “dysautonomic” phenotype [10].

Phenotypically, PD-related hyperhidrosis may be generalized or localized, symmetric or asymmetric, and either persistent or fluctuating. Fluctuating forms are most often tied to dopaminergic “off” states or peak-dose dyskinesia [64,65]. A large-scale thermal sweat testing study of 225 PD patients documented hyperhidrosis in 14.7%, split between generalized hyperhidrosis (GHH), which refers to excessive sweating that occurs across large areas of the body, and compensatory hyperhidrosis (CHH), which is characterized by excessive sweating that develops in new regions; CHH often co-occurs with distal hypohidrosis. Although both subtypes have similar sweat thresholds, GHH shows significantly higher leg sweat rates (0.173 ± 0.085 mg/cm^2^/min vs. 0.054 ± 0.049 mg/cm^2^/min; *p* = 0.0001) [56]. Fluctuating hyperhidrosis can also emerge during dopaminergic withdrawal syndromes or during pregnancy in PD [49,64].

Management of hyperhidrosis is challenging, given the paucity of PD-specific trials. First-line measures focus on environmental modification (cool ambient temperatures, breathable fabrics), hydration optimization, and dopaminergic regimen adjustment to reduce non-motor fluctuations [49]. Topical aluminum salts, anticholinergics (oral or topical glycopyrrolate), and botulinum toxin are effective in primary hyperhidrosis but lack robust PD-specific validation [63]. Systemic anticholinergics require caution in older patients due to potential cognitive adverse effects. For refractory cases, botulinum toxin injections targeting high-burden areas (e.g., palmar, plantar, and axillary) may be considered [66,67]. For patients who are candidates for DBS surgery, the effects of stimulation parameters on hyperhidrosis may be considered [68]. A prospective study of 60 PD patients undergoing subthalamic nucleus (STN) DBS followed the patients up to a year after surgery and reported significant and lasting positive effects on thermoregulation, documented via the NMS questionnaire [69]. Another study involving 19 PD patients reported thatdyshidrosis was reduced by 66.7% at 6 months post- STN DBS [68,70].

### 4.2. Anhidrosis, Hypohidrosis, and Heat Intolerance

Hypohidrosis (reduced sweating) and anhidrosis (complete absence of sweating) are frequent and clinically meaningful manifestations of thermoregulatory dysfunction in PD. Although prevalence estimates vary with methodology, site selection, and disease stage, objective sudomotor testing consistently demonstrates reduced sweat output [15,48,49]. Patient-reported outcomes further show frequent heat intolerance and activity limitations in warm conditions, aligning real-world disability with central–peripheral autonomic changes [71].

Quantitative Sudomotor Axon Reflex Test (QSART) studies consistently reveal a distal-predominant postganglionic pattern, characterized by significant sweat volume reductions in the feet and distal legs, even in early stages of the disease, with progressive decline correlating with the Hoehn/Yahr stage [15,16]. Longitudinal use of the thermoregulatory sweat test (TST)—which evaluates whole-body sweat output in response to a thermal stimulus, in contrast to the QSART, which measures localized postganglionic axon reflex sweating—and the QSART in centers with the proper equipment and expertise can document disease-related progression and guide counseling on environmental and behavioral adaptations. However, the need for proper equipment and experts for conduct and interpretation of the TST limit its usage [52]. In a two-year longitudinal study, PD patients exhibited measurable deterioration in sudomotor function over time, especially at lower-limb sites, underscoring the progressive nature of sweat gland hypofunction [52]. Reduced sweating in these regions contributes to an impaired ability to dissipate heat during environmental or exertional thermal loads, predisposing patients to heat intolerance, and, in extreme cases, hyperthermia. When used properly, QSART can show disease related progression and guide counseling efforts. Clinically, anhidrosis and hypohidrosis may present insidiously, with patients reporting difficulty tolerating heat, flushing, lightheadedness, or disproportionate fatigue during warm weather or physical activity [48]. Objective testing frequently reveals subclinical deficits, and their extent can be underestimated by protocols restricted to proximal sites [31,32].

The functional consequences of impaired or absent sweating are significant: heat intolerance can limit participation in outdoor activities, exercise programs, and social events, contributing to social isolation and reduced quality of life. During heat waves or febrile illness, these deficits may place patients at elevated risk for dangerous hyperthermic states, particularly when compounded by other autonomic impairments such as orthostatic hypotension [17,52]. Medication effects may exacerbate these risks. Anticholinergic agents are sometimes used in the management of PD (e.g., trihexyphenidyl for tremor, glycopyrrolate for drooling, or oxybutynin for neurogenic bladder), but each may reduce sweating further [72]. “Off” states may precipitate sweating abnormalities in PD, with patients reporting episodes that cluster in off periods. Dyskinesia also may provoke abnormal sweating, although to a lesser extent. Patterns of increased sweating may be generalized or regional and can be distressing [48]. Management strategies are aimed primarily at symptom mitigation and prevention of heat-related morbidity. Patient education focuses on maintaining cool environments, ensuring adequate hydration, using evaporative cooling devices, and scheduling activity during cooler times of the day. For those with combined hypohidrosis and compensatory hyperhidrosis, targeted interventions may be needed to address both the functional heat dissipation deficit and the quality-of-life impact of excessive sweating in unaffected regions. Further research is needed to define whether early identification and intervention for hypohidrosis can mitigate downstream risks of hyperthermia and associated complications in PD.

### 4.3. Hyperthermia and Parkinsonism Hyperpyrexia

Hyperthermia in PD arises from both intrinsic thermoregulatory failure and secondary systemic or iatrogenic triggers. Neurodegeneration within the hypothalamus, brainstem, and intermediolateral spinal cord disrupts central heat-regulating pathways, while peripheral sudomotor and vasomotor dysfunction reduces evaporative and convective heat loss. This combination, compounded by bradykinesia, rigidity, and reduced mobility, limits the ability of patients to seek cooling and avoid excessive warming, increasing susceptibility to heat stress. Environmental temperature elevations can precipitate disproportionate increases in core temperature (up to 41 °C in some case reports), particularly in patients with hypohidrosis or anhidrosis, leading in some cases to heat exhaustion or frank hyperthermia [73].

Parkinsonism–Hyperpyrexia Syndrome (PHS) is one of the most severe and life-threatening hyperthermic manifestation in PD [74,75]. PHS is a hypodopaminergic crisis most frequently precipitated by abrupt withdrawal, dose reduction, or gastrointestinal (GI) malabsorption of dopaminergic medications. Other recognized triggers include systemic infection, dehydration, surgery, and metabolic stress [74,76]. Across published series and case reports, onset typically occurs within hours to 7 days of dopaminergic interruption. Clinically, PHS is characterized by sustained hyperpyrexia, frequently exceeding 40 °C, severe generalized rigidity, altered mental status (from confusion to coma), and autonomic instability with labile blood pressure and heart rate. Laboratory findings include leukocytosis (often >12,000/μL), elevated creatine kinase (CK frequently >1000 U/L, with reports exceeding 10,000 U/L in cases with severe rhabdomyolysis), and, in some cases, metabolic acidosis or elevated transaminases. Documented complications include acute renal failure from myoglobinuric rhabdomyolysis, aspiration pneumonia, venous thromboembolism with pulmonary embolism, disseminated intravascular coagulation, and prolonged neurologic deficits [74,77]. Mortality rates of 5–20% have been reported, with higher fatality in elderly, frail, or late-diagnosis cases. Perioperative settings represent a high-risk context for PHS. Several reports describe fulminant onset following perioperative withholding of levodopa, delayed postoperative resumption, or ineffective absorption due to postoperative ileus [78,79]. In one perioperative case, symptoms developed within 24 h of the last preoperative dose, with temperature peaking at 41 °C, CK rising to 8500 U/L, and subsequent acute renal injury. Such cases may initially be misdiagnosed as malignant hyperthermia; however, unlike malignant hyperthermia, PHS does not respond to dantrolene, and definitive management requires rapid reinstatement of dopaminergic therapy.

A hyperdopaminergic variant, dyskinesia-hyperpyrexia syndrome (DHS), has also been described, triggered by levodopa overdose or impaired metabolism [80]. DHS presents with severe continuous dyskinesia, hyperpyrexia, elevated CK, and rhabdomyolysis. Prevention of such cases in PD is two-pronged: anticholinergic/sudomotor-impairing burden during hot weather or febrile illness must be avoided and prevention of dopaminergic withdrawal (peri-procedural plans, treatment of malabsorption) that can precipitate hyperthermic crises is vital [81,82,83].

Device-related causes are also documented. DBS-related PHS can occur after abrupt cessation of deep brain stimulation due to battery depletion, lead fracture, programming error, or perioperative deactivation, even without medication withdrawal [84,85]. Emergence of rigidity and dyskinesia symptoms that were under control with DBS stimulation may result in symptoms of PHS in such cases.

Management of PHS, whether medication- or DBS-related, centers on immediate restoration of dopaminergic tone, preferably via the patient’s usual regimen of oral intake if possible, or via nasogastric, transdermal, or subcutaneous routes otherwise. Aggressive supportive measures are critical: hydration, active external cooling, empiric antibiotics when infection is suspected, thromboprophylaxis, and close monitoring for systemic complications [74,76]. In DBS-related PHS, urgent reactivation of stimulation or emergent hardware replacement can be life-saving, with some reports documenting resolution of hyperthermia and rigidity within hours of stimulation restoration [85,86,87]. Preventive strategies include maintaining uninterrupted dopaminergic therapy during hospitalizations, implementing perioperative medication continuity protocols, ensuring timely DBS battery replacement, and educating patients and caregivers to recognize early warning signs. Because PHS can be clinically indistinguishable from neuroleptic malignant syndrome (NMS) or serotonin syndrome, and because delayed treatment worsens prognosis, a high index of suspicion and early dopaminergic rescue are essential for favorable outcomes [73,77].

### 4.4. Hypothermia

Hypothermia in PD is rare but potentially life-threatening. It is the consequence of combined hypothalamic, brainstem, spinal, and peripheral autonomic degeneration that impairs both heat production and conservation [50]. In a 30-patient series of accidental hypothermia in PD, occurring most frequently during winter months, mean core temperatures at hospital admission ranged from 30.0 to 33.5 °C and were preceded in 83% of cases by several days of worsening bradykinesia, limb coldness, or confusion [11]. Cardiac ^123^I-metaiodobenzylguanidine (MIBG) imaging showed markedly reduced heart-to-mediastinum uptake ratios (mean 1.18 ± 0.22), consistent with severe sympathetic denervation. Electrocardiographic findings included bradycardia and Osborn J waves in 43% of patients. Recovery typically occurred within 12–48 h. after active rewarming, with no deaths reported [11,88]. Dopamine agonists, particularly pramipexole, were implicated in 30% of cases, possibly by blunting thermogenic responses to cold exposure, though hypothermia also occurred in drug-naïve patients [11]. This may represent a class effect, as other case reports have linked bromocriptine and lisuride to hypothermia [89]. Curiously, psychiatric inpatients exposed to dopamine antagonists have also shown higher rates of hypothermia (2.6% vs. 1.3% non-exposed, *p* = 0.024) [90].

Because TI is initiated by inhibition of the POA it is reasonable to hypothesize that, in PD, hypothalamic neuronal degeneration could give rise to a paradoxical, inverted thermoregulatory paradigm [35]. This may help explain why hypothermia appears in these patients most frequently during the winter months. In this scenario, rather than a simple disruption of thermogenic capacity, there may be an active inhibition of thermogenesis triggered by cold exposure. If true, the TI mechanism could also account for why PD patients exposed to high ambient temperatures are more prone to developing hyperthermia. However, more studies are needed to explore this possibility. Spontaneous periodic hypothermia (SPH) has also been described in PD with hypothalamic α-synuclein pathology, producing recurrent episodes of hypothermia (31–33 °C) accompanied by bradycardia, hypotension, hypoglycemia, and altered consciousness. These episodes typically resolve completely after initiation of carbidopa/levodopa, highlighting that dopaminergic therapy can restore hypothalamic temperature regulation, with effects appearing in an essentially on–off pattern, rather than one of graded modulation [88]. Together, these data indicate that PD-related hypothermia is multifactorial, often seasonal, frequently preceded by prodromal motor/autonomic changes, and reversible with prompt rewarming and dopaminergic optimization. It also supports the idea of unbalanced dopaminergic tone or the influence of underlying synuclein pathology on differential responses.

### 4.5. Medication-Related Thermoregulatory Effects

Drug exposures can unmask or amplify thermoregulatory vulnerability in PD via several mechanisms: reduced sweating (anticholinergics and other anhidrotic drugs), impaired skin blood flow responses (some cardiovascular agents), hypodopaminergic crises (withdrawal or “off” states), and paradoxical dopaminergic effects, each producing clinically important hyperthermia or hypothermia phenotypes [81]. In controlled heat stress, medications with high anticholinergic activity raise core temperature and blunt sweating: at ambient temperatures ≥ 30 °C, agents with high anticholinergic properties (e.g., atropine) increased core temperature by +0.42 °C (95% CI 0.04–0.79) with concomitant reductions in sweat rate [82]. Nonselective βblockers (e.g., propranolol) were associated with a smaller but measurable rise (+0.11 °C, 95% CI 0.02–0.20) under heat stress [82]. Real world pharmacovigilance signals reinforce these experimental findings: analysis of the WHO VigiBase (2000–2020) identified 368 reports- of drug induced hypohidrosis and 128 of anhidrosis, with disproportionate reporting linked to nervous system drugs and anticholinergics (with specific mentions of oxybutynin, topiramate, and zonisamide). Impairment of sweating provides a physiologic basis for heat intolerance and hyperthermia when behavioral and autonomic cooling responses would normally be active [83]. In PD specifically, a case report documents oxybutynin-associated hyperthermia attributed to antimuscarinic suppression of sweating, underscoring caution when treating urinary symptoms in heat vulnerable patients [72]. Newer anticholinergics, such as solifenacin, darifenacin, and imidafenacin, are more selective M3 acetylcholine receptor antagonists and thus exhibit higher uroselectivity and exert less disruption of sweating and core temperature control than older, nonselective agents [91]. Beyond anticholinergics, PD “OFF” physiology may interfere with thermoregulation. In a physiologic study using evaporimetry, a test measuring the rate of sweat evaporation on three different areas of the body (right hand, left hand, chest) immediately before the morning PD medication (baseline), and hourly thereafter for 4 h, individuals with wearing off had significantly increased sweating in the initially affected hand during the OFF state (with *p*-values < 0.05 across hands), whereas patients without wearing off showed no such fluctuation [92]. Clinically, this aligns with survey data showing frequent and burdensome OFF-related non-motor symptoms (including autonomic complaints), although heat stress metrics themselves are best captured in experimental and pharmacovigilance datasets [82,83].

Broader reviews of drug-induced hyperthermia describe overlapping management principles that include rapid identification, active cooling, aggressive supportive care, and syndrome-specific therapy (e.g., reestablishing dopaminergic -stimulation for PD hypodopaminergic states; benzodiazepines/cyproheptadine for serotonin syndrome), reinforcing that recognition of the medication mechanism guides rescue [81]. Medication mechanisms relevant to hyperthermia (OFF/PHS) and hypothermia (antipsychotics, dopamine agonists) are detailed in the syndrome-specific sections above.

A patient’s functional level may interact with medications and produce thermoregulatory dysfunction (e.g., OFF-related fluctuations and polypharmacy heighten risk). Contemporary practice guidance also emphasizes anticholinergic minimization in older adults with multimorbidity and cognitive vulnerability [13]. PD care teams should (i) screen for heat intolerance- and sweating changes when starting or escalating anticholinergics and specific CNS agents linked to anhidrosis/hypohidrosis [82,83], (ii) educate patients to help them adjust behavior during heat waves and intercurrent illness [81,82], (iii) plan perioperative dopaminergic continuity to avoid hypodopaminergic hyperthermia [81,93], and (iv) recognize and treat medication-linked hypothermia—particularly with antipsychotics or dopamine agonists—through dose review, drug withdrawal when appropriate, warming, and monitoring [89,90].

## 5. Diagnostics and Tests

### 5.1. Quantitative Sudomotor Axon Reflex Test (QSART)

The QSART quantifies postganglionic sudomotor integrity by iontophoresing acetylcholine over selected skin sites to trigger an antidromic axon reflex with ensuing orthodromic activation of neighboring sweat glands. Normal sweat volume denotes intact postganglionic fibers, whereas reduced or absent output indicates sudomotor axonopathy [51,94]. In PD, abnormalities typically follow a distal, length-dependent pattern, and studies focused only on proximal or single sites tend to underestimate involvement; greater decrements in leg sweat volumes are seen with advancing Hoehn/Yahr stage, consistent with progression of the disease as well as autonomic dysfunction [18,94,95]. Additional studies have confirmed QSART’s ability to detect small-fiber sudomotor deficits in PD. Deficits are most prominent in the feet and distal legs, even in early disease, and correlate with both composite autonomic severity scores and other thermoregulatory measures [17]. QSART reproducibly identifies progressive sweat volume decline over time in PD, particularly in lower limb sites, supporting its use as a longitudinal biomarker.

Pathology lines up with the physiology: skin immunofluorescence shows a length-dependent gradient of phosphorylated α-synuclein in Parkinson’s disease, with greater distal than proximal deposition that mirrors the distal-predominant QSART deficit; related work also points to distinct topographies across synucleinopathies [96]. In a large multicenter, blinded study, skin biopsy detected phosphorylated α-synuclein in the vast majority of clinically confirmed cases (≈93% in PD and ≥96–100% in atypical cases characterized by synuclein pathology), and the quantitative burden correlated with disease severity measures, supporting clinical relevance [97]. Together, these cutaneous biomarkers complement QSART by localizing a postganglionic lesion that explains PD’s distal-greater-than-proximal sweat loss and by offering a practical, repeatable tissue assay for longitudinal tracking [96,97].

QSART abnormalities, when interpreted alongside the Thermoregulatory Sweat Test (TST), can aid in differentiating postganglionic from preganglionic lesions. The autonomic deficit in PD is primarily postganglionic, in contrast with MSA, where the deficit is predominantly, though not exclusively, preganglionic. Comparative studies using the autonomic reflex screen, a non-invasive battery consisting of QSART, Valsalva analysis, heart rate variability to deep breathing, and a tilt table test, show that PD patients generally have milder Composite Autonomic Severity Score (CASS) subscores than MSA patients (the CASS is a 10-point scoring system that quantifies autonomic nervous system dysfunction across sudomotor, cardiovagal, and adrenergic domains). In practice, its quantitative readout, sensitivity to small-fiber disease, and established uses explain its popularity and continued role as a gold-standard measure in both research and routine care. Its usefulness in differential diagnosis and assessment of severity make QSART an essential tool [98,99].

### 5.2. Thermoregulatory Sweat Test (TST)

The TST interrogates the entire efferent thermoregulatory pathway by coating the anterior skin surface with an alizarin-red indicator and inducing controlled heat stress in a temperature- and humidity-regulated chamber, after which percent anterior body anhidrosis and the topography of sweat/anhidrosis are quantified [100,101]. When interpreted alongside QSART, lesion site can be inferred: absent sweating on TST with normal QSART indicates a preganglionic defect, whereas concordant QSART reduction implies postganglionic involvement; in PD, the combined pattern frequently reflects length-dependent anhidrosis with a predominantly postganglionic deficit [51]. In PD, TST can track changes over time, with one cohort showing a rise in mean anhidrosis from about 10% to 14% over a year; in contrast, MSA patients began near 57% and increased by roughly 6% annually [51,95,102]. Compared with simpler methods, such as the Sympathetic Skin Response (SSR) test, which offers easier administration but lower sensitivity for early autonomic changes, TST provides higher specificity and detailed lesion mapping, albeit with greater logistical demands [99,103]. For differential diagnosis within the synucleinopathies, dementia with Lewy bodies typically shows intermediate values and a distal pattern akin to PD, whereas MSA exhibits a regional or generalized pattern with acral sparing; thus, a high burden, generalized pattern on TST should prompt consideration of MSA rather than PD. It is important to note that, while this characteristic abnormality on TST can be viewed as a non-motor feature that supports the diagnosis of MSA, core clinical motor and non-motor symptoms should remain the mainstay of diagnosis [104]. In practice, these characteristic PD features, low overall anhidrosis with distal distribution, make TST especially useful to corroborate postganglionic sudomotor involvement and to provide an objective benchmark for change over time when available [51].

### 5.3. Sympathetic Skin Response (SSR)

The SSR records transient electrodermal potentials from palms/soles in response to suprathreshold stimuli (e.g., deep breathing or electrical pulses), with efferent pre- and postganglionic sudomotor pathways to the skin [51,105,106]. It is well-established, rapid, and widely deployed for screening (including historically in lie-detection) because of its sensitivity to phasic sudomotor activity, but its clinical specificity is limited by its high inter- and intra-subject variability, habituation, and non-localizing nature [99]. In a comparative study that enrolled 61 neurological patients and 50 age-matched healthy controls, SSR amplitudes were preserved in participants with vasovagal syncope (*n* = 25) and PD (*n* = 15), whereas mean palm/sole SSR amplitudes were significantly reduced in MSA (*n* = 11) and peripheral neuropathy (PN; *n* = 10) [103]. Thus, SSR is popular as a screening adjunct but requires confirmatory testing for lesion localization and diagnosis. In PD specifically, SSRs are often preserved and may not differ from age-matched controls, so a normal SSR does not exclude PD-related sudomotor involvement. By contrast, atypical parkinsonism and PN show more frequent loss, with absent palm SSR in 6/11 (≈55%) MSA and 4/10 (40%) PN, and absent sole SSR in 5/11 (≈46%) MSA and 5/10 (50%) PN, with significantly reduced mean amplitudes versus controls. Thus, an absent or markedly reduced SSR in a parkinsonian patient should prompt evaluation for MSA or concomitant neuropathy [99,103].

### 5.4. Electrochemical Skin Conductance (ESC; Sudoscan)

ESC uses reverse iontophoresis on stainless-steel electrodes under hands and feet to measure electrochemical skin conductance driven largely by chloride ion flux, yielding palm (P-ESC) and sole (S-ESC) values that index sudomotor function without localization [103]. Compared with SSR as a reference for abnormal sudomotor function, ESC shows high diagnostic accuracy: sensitivity 0.91 (P-ESC) and 0.95 (S-ESC), specificity 0.78 and 0.85 [103]. While attractive for speed and ease in general clinics, ESC does not identify lesion level and has reported low sensitivity for sweat-gland fiber loss in some contexts, so it complements rather than replaces QSART/TST in comprehensive evaluations but can be a useful substitution if QSART is not readily available [107]. In cross-sectional comparisons across autonomic disorders, PD patients’ palm/sole ESC values frequently overlap with age-matched controls—whereas MSA and PN show clear abnormalities. Therefore, a normal ESC does not exclude PD-related sudomotor involvement, and results should be interpreted alongside symptom scales and, when feasible, QSART/TST [103].

### 5.5. Laser Doppler Flowmetry (LDF) and Imaging (LDI) of the Vasomotor Axon Reflex

LDF/LDI quantifies the neurogenic vasodilatory “flare” elicited by cholinergic iontophoresis as a measure of nociceptive C-fiber integrity. LDF samples a single point with high temporal resolution, whereas LDI maps a two-dimensional perfusion field with spatial and temporal resolution [108]. LDF suffers from high intra- and inter-individual variability that undermines diagnostic use, whereas LDI reduces variability via spatial averaging but still lacks standardized image-analysis methods; therefore, clinical deployment remains limited or experimental in most centers [108]. Compounding this, iontophoresis protocols vary in current density, dose, and acetylcholine concentration across studies, making direct comparisons difficult and reinforcing the need for methodological harmonization before broader clinical uptake. Once these technical limitations are resolved, this tool may have potential for assessing small fiber neuropathies, monitoring autonomic dysfunction, and evaluating therapeutic responses in peripheral nerve disorders.

### 5.6. Sensitive Sweat Test and Spoon Test

The sensitive sweat test uses dye-impregnated papers to detect very low sweat rates, enhancing early detection sensitivity relative to cruder methods, whereas the Spoon Test is a bedside maneuver in which sweat adherence to a warmed metal spoon qualitatively signals sudomotor activity; Although simple and resource efficient, both provide only semi-quantitative or qualitative information and thus serve as screening tests, to be followed by quantitative testing when abnormal [99].

### 5.7. Experienced Temperature Sensitivity and Regulation Survey (ETSRS)

The ETSRS is a 134-item, online patient-reported survey designed to study thermoregulation and thermosensitivity in research settings [109]. It includes 102 questions about thermal experiences, which document an individual’s subjectively experienced sensitivity to temperature and their thermoregulatory responses (e.g., “Compared to others, do you feel cold more quickly in winter?”). It also asks 32 questions addressing how these experiences vary across body locations, thereby providing a detailed map of perceived temperature sensitivity [e.g., “Do you feel cold more strongly in your hands and feet than in other body parts?”). Factor analysis has identified 14 independent components, including a “Heat & Activity-Induced Autonomic Thermoregulation” domain that captures thirst and sweating responses, with a median completion time of approximately 9.5 min [109]. Widely used PD questionnaires, such as the Scale for Outcomes in Parkinson’s disease-Autonomic Dysfunction (SCOPA-AUT) and the Non-Motor Symptoms Questionnaire (NMSQ), contain very few questions regarding thermoregulation or sudomotor symptoms, and objective gold-standard tests (QSART, TST) are technically demanding and concentrated in specialty centers [51,99,103]. ETSRS could fill a practical gap by quantifying heat, activity, and stress-evoked sweating across time of day and seasons and by localizing symptoms. In this context, ETSRS can complement traditional approaches by identifying patient-reported patterns of temperature sensitivity and aligning them with physiological findings, though formal validation in PD cohorts remains necessary [109]. Different tests and diagnostics serve different clinical purposes and vary in usage as shown in Table 1. 

### 5.8. Emerging and Research-Based Tests

Several novel methods are being explored to assess sudomotor and related autonomic functions, though all remain primarily research tools rather than routine clinical diagnostics. Silicone-impression mapping and the related quantitative direct and indirect reflex testing (QDIRT) provide high-resolution visualization of sweat droplet activity and temporal patterns, offering detailed mapping at low cost, but are still limited by the lack of normative datasets and automation [99]. The Quantitative Pilomotor Axon Reflex Test (QPART) assesses pilomotor (goosebump) responses to pharmacologic stimulation and may help evaluate small-fiber integrity, but its role is still experimental and requires larger studies before clinical adoption [110]. Microfluidic patches represent a wearable approach for continuous sweat analysis, capable of detecting sweat rate and biochemical markers such as pH, chloride, and even levodopa levels, though they remain at the proof-of-concept stage [111]. Together, these technologies hold promise for more accessible or detailed autonomic testing, but at present they are best viewed as investigational complements to established tools such as QSART and TST.

Comparison of commonly used and bedside tests: QSART (Quantitative Sudomotor Axon Reflex Test), TST (Thermoregulatory Sweat Test), SSR (Sympathetic Skin Response), ESC (Electrochemical Skin Conductance; Sudoscan device), LDF/LDI (Laser Doppler Flowmetry/Laser Doppler Imaging), and Sensitive Sweat/Spoon test (qualitative, low-resource screening). Rows summarize target pathway measured; sensitivity/specificity notes (including mentions of PD = Parkinson’s disease, MSA = Multiple System Atrophy, PN = peripheral neuropathy); whether the test is lesion-localizing; laboratory requirements; specialized equipment; trained personnel; estimated cost; and key utility notes.

## 6. Management of Thermoregulatory Dysfunction in Parkinson’s Disease

### 6.1. Hyperhidrosis

Environmental control remains the mainstay of hyperhidrosis management in PD. Patients benefit from maintaining cool indoor environments, wearing breathable fabrics, and avoiding thermal stress [12]. Limited data suggests that STN DBS may improve autonomic stability and result in a significant reduction in sweating dysfunction and improved skin sympathetic responses [112]. Smaller cohorts confirm improvements in both sleep and excessive sweating after DBS [69], and electrophysiological studies demonstrate normalization of sympathetic skin responses [113]. It is possible that the improvement in these non-motor symptoms is due to better management of fluctuations through stimulation. It is important to note, however, that thermoregulatory dysfunction is not considered a first-line indication for DBS and should not be the main goal for consideration of this surgery.

Topical aluminum salts and glycopyrrolate are borrowed from primary hyperhidrosis treatment, though evidence in PD is limited and systemic anticholinergics risk cognitive adverse effects in older adults [12,13]. Botulinum toxin injections may provide targeted relief in refractory cases. Optimization of dopaminergic regimens to reduce OFF periods is crucial, since sweating abnormalities are strongly linked to motor fluctuations [92]. Generalized hyperhidrosis may also emerge spontaneously in PD, independent of dyskinesias or heat exposure, likely reflecting underlying autonomic dysregulation that disrupts normal sympathetic control of sweating. A summary of the pharmacologic and non-pharmacologic measures used in the management of hyperhidrosis can be found in Table 2.

### 6.2. Hypothermia

Seasonal hypothermia in can occur in PD. External rewarming through heated blankets, warmed IV fluids, and climate control can restore normothermia within 12–48 h [11]. Caregiver education on early signs such as bradykinesia and confusion is critical [88]. Awareness of the complex influences of dopaminergic therapy is important. Clinicians should gather a detailed medication list with timing information to determine appropriate next steps. A summary of the pharmacologic and non-pharmacologic measures used in the management of hypothermia can be found in Table 3.

### 6.3. Anhidrosis, Hypohidrosis, and Heat Intolerance

Management of reduced sweating in PD focuses on preventive and supportive strategies to mitigate heat-related morbidity. Non-pharmacologic measures are the cornerstone of care. Patients should be advised to avoid environmental heat exposure through use of air conditioning, climate-controlled environments, and activity scheduling during cooler times of day [12]. Evaporative cooling devices, such as misting sprays, handheld fans, and cooling vests, provide external substitutes for impaired sweating [12]. Hydration protocols are critical, as dehydration amplifies heat intolerance [49]. Patient and caregiver education is essential for recognizing early signs of heat stress and implementing rapid cooling interventions. Climate change projections reinforce the importance of anticipatory adaptation strategies, including access to cooling centers during heatwaves [14].

Pharmacologic management focuses primarily on avoiding drugs that exacerbate sudomotor dysfunction. Anticholinergic medications are a major concern [83]. Consensus updates recommend minimizing anticholinergic burden in older adults, especially those with cognitive impairment or polypharmacy [13]. Adjustments to dopaminergic therapy may also be beneficial, as OFF-related fluctuations exacerbate sweating abnormalities; studies show sweating increases in the affected hand during wearing-off states [92]. Other drugs, such as non-selective β-blockers, may further elevate heat stress risk by attenuating heat dissipation, suggesting careful review of comorbid medications [82]. A summary of the pharmacologic and non-pharmacologic measures used in the management of anhidrosis, hypohidrosis, and heat intolerance can be found in Table 4.

### 6.4. Hyperthermia and Parkinsonism–Hyperpyrexia Syndrome

Environmental adaptation is critical: cooling garments, hydration, and avoidance of heat waves are first-line preventive strategies [14]. Second, uninterrupted dopaminergic therapy is essential, and careful planning during inpatient stays and/or DBS battery replacements are important to prevent PHS [12,84].

Rapid reinstatement of dopaminergic therapy is the definitive treatment for PHS. Levodopa or dopamine agonists delivered via nasogastric, transdermal, or subcutaneous routes to reverse rigidity, autonomic instability, and hyperthermia should be prioritized. In contrast to neuroleptic malignant syndrome, antipyretics and dantrolene are ineffective in PHS, underscoring the importance of dopaminergic rescue. Preventive strategies include strict medication continuity and timely DBS maintenance [69,112]. This is especially important to prevent PHS during hospitalization for PD patients. A summary of the pharmacologic and non-pharmacologic measures used in the management of hyperthermia and PHS can be found in Table 5.

### 6.5. Medication-Related Thermoregulatory Effects

Education on hydration, heat avoidance, and early recognition of temperature dysregulation is emphasized for patients and caregivers, particularly when anticholinergic or other thermoregulatory-impairing agents are prescribed [13].

Anticholinergics suppress sweating and can precipitate hyperthermia, with oxybutynin-associated cases reported in PD [72]. One meta-analysis shows anticholinergic burden raises core temperature during heat stress [82], and WHO pharmacovigilance data identify hundreds of reports of drug-induced anhidrosis linked to anticholinergics and CNS drugs [83]. Abrupt dopaminergic withdrawal can trigger lethal hyperthermia, as in “on–off” crises with temperatures exceeding 41 °C [93]. Dopamine agonists can conversely induce paradoxical hypothermia, including severe bromocriptine-induced episodes [89]. Antipsychotics further elevate hypothermia risk, doubling prevalence in psychogeriatric inpatients [90]. Clinical guidance emphasizes minimizing anticholinergic exposure and tailoring drug regimens to reduce thermoregulatory risk [51]. Although early animal studies show that amantadine can induce centrally mediated, dopamine-independent hypothermia in rodents, comparable effects have not been demonstrated in humans, and its relevance to thermoregulation in PD remains unclear [114]. A summary of the pharmacologic and non-pharmacologic measures used in the management of medication-related thermoregulatory effects can be found in Table 6.

## 7. Gaps and Areas for Future Research

Thermoregulatory dysfunction in PD spans the full spectrum—from hyperhidrosis to hypohidrosis and anhidrosis—and frequently includes heat and cold intolerance that erode quality of life and can lead to social isolation. Management is fundamentally preventive: optimize dopaminergic therapy to limit OFF-related sweating swings; minimize or avoid drugs that impair sweating or cutaneous vasodilation (especially anticholinergics); and pair medication review with practical measures such as cool environments, hydration, and targeted cooling strategies. Perioperative and device management should ensure uninterrupted dopaminergic delivery to avert hyperthermic crises. Looking ahead, the field needs PD-specific treatment trials for both hyperhidrosis and hypohidrosis, scalable monitoring outside specialty centers (wearables and robust patient-reported metrics), clearer phenotyping that links thermoregulatory profiles to motor and non-motor subtypes and autonomic biomarkers, and consensus guidance for perioperative care and heat-wave preparedness. Future investigations should also explore how PD alters BAT thermogenesis and shivering responses, as understanding the interplay between these two heat-generating mechanisms may reveal novel insights into impaired cold defense and metabolic regulation in PD patients. A proactive, prevention-focused approach, coupled with these research advances, offers the best path to reduce morbidity and improve daily participation and safety for people living with PD. Future work should also address major knowledge gaps in how medications influence thermoregulatory function and evaluate these responses across atypical parkinsonian disorders, where differing patterns of autonomic involvement may alter therapeutic needs and outcomes.

## 8. Conclusions

Thermoregulatory dysfunction in Parkinson’s disease reflects a complex interplay between central hypothalamic–brainstem degeneration, peripheral autonomic failure, and medication effects. This multisystem disruption compromises sweating, vasomotion, and thermogenesis, leading to intolerance of environmental extremes and, in severe cases, life-threatening hyper- or hypothermic crises. Clinicians should recognize these manifestations as integral to PD’s non-motor spectrum and adopt preventive, patient-centered strategies emphasizing dopaminergic stability, environmental adaptation, and avoidance of anticholinergic burden. At the clinical research level, integrating autonomic biomarkers with wearable and imaging-based assessments could clarify pathophysiologic subtypes and inform targeted interventions. Finally, it is crucial to also continue investigating autonomic and thermoregulatory networks in animal models, which remain the only viable approach for studying these mechanisms in depth and for advancing treatments for specific pathophysiological autonomic dysfunctions in humans.

Investigation into altered BAT activation and impaired shivering thermogenesis in PD may open new avenues for understanding and mitigating cold defense deficits, ultimately improving thermoregulatory resilience and patient outcomes.

## Figures and Tables

**Figure 1 cells-14-01910-f001:**
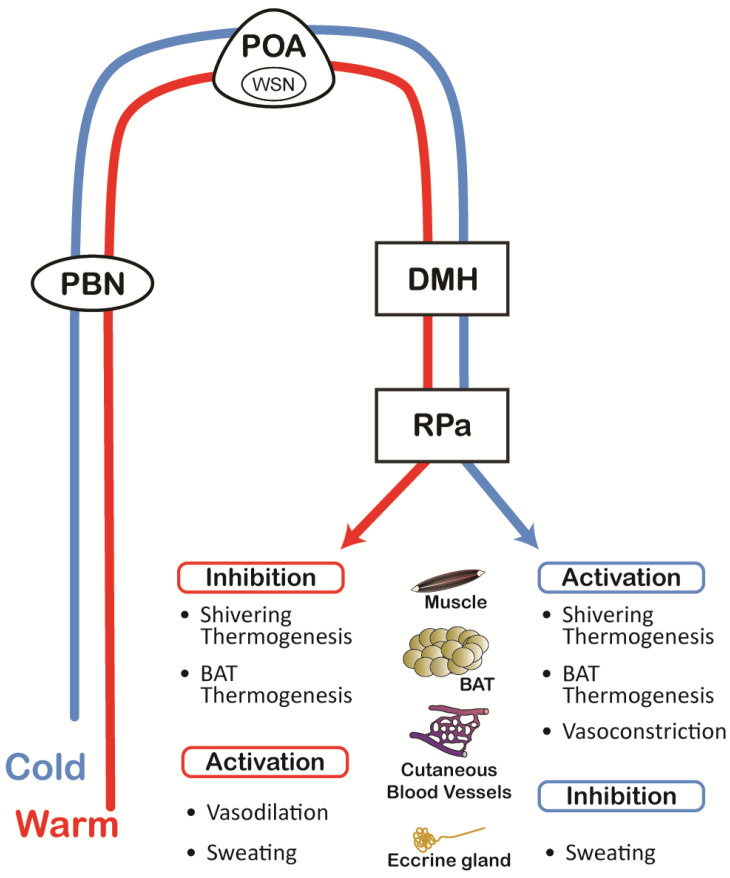
Normal Thermoregulatory Pathway.

**Table 1 cells-14-01910-t001:** Summary of Tests.

	*QSART*	*TST*	*SSR*	*ESC (Sudoscan)*	*LDF/LDI*	*Sensitive Sweat/Spoon Test*
*Measures/Target Pathway*	Postganglionic sudomotor (ACh iontophoresis)	Entire efferent thermoregulatory pathway	Central + pre-/postganglionic sudomotor	Electrochemical skin conductance (reverse iontophoresis); composite sudomotor index; non-localizing.	Vasomotor C-fiber axon reflex	Low sweat detection (qualitative)
*Sensitivity/Specificity*	High; variable by site	High	Low–moderate for PD; amplitudes often preserved in PD but reduced/absent in MSA/PN; non-localizing.	High for MSA/PN; lower for PD	Moderate; research-based	Low–Moderate
*Lesion Localizing*	Yes (postganglionic)	Yes (with QSART)	No	No	No	No
*Lab Required*	Yes	Yes	No (basic EMG)	No	Yes (research)	No
*Specialized Equipment*	QSART device, climate control	Heat chamber, imaging	EMG machine	ESC device	Laser Doppler	Minimal
*Trained Personnel Needed*	Yes	Yes	Moderate	Minimal	Yes	Minimal
*Estimated* *Cost*	High	High	Low	Moderate	High	Very Low
*Utility Notes*	Sensitive to distal PD patterns; tracks progression	Distinguishes PD (low % anhidrosis) from MSA (high %)	Often normal in PD; useful as a rapid screen; absent or markedly reduced responses should prompt evaluation for MSA or neuropathy	Fast PD screen; normal results don’t exclude dysfunction; interpret with clinical context/QSART-TST.	Limited clinical PD data; mostly investigational	Quick screen in low-resource PD settings

**Table 2 cells-14-01910-t002:** Management of Hyperhidrosis.

Non-Pharmacologic Measures	Pharmacologic Measures	Clinical Notes
Cool indoor environments, breathable fabrics, and avoidance of thermal stress. Schedule exercise in cooler parts of the day; use portable fans, cooling towels, or cold packs for symptom relief. Educate patients on early recognition of overheating and safe cooling practices.	Optimize dopaminergic regimen to reduce OFF-related hyperhidrosis and autonomic swings (maintain adherence, adjust timing, improve GI absorption). Topical aluminum chloride or glycopyrrolate for focal sweating. Botulinum toxin for refractory focal hyperhidrosis.	Prefer topical/local options; avoid systemic anticholinergics in older adults (see separate medication table). Adjust dopaminergic therapy to prevent abrupt withdrawal and lethal hyperthermia.

**Table 3 cells-14-01910-t003:** Management of Hypothermia.

Non-Pharmacologic Measures	Pharmacologic Measures	Clinical Notes
Seasonal prevention: warm indoor environment, layered breathable/moisture-wicking clothing, heated bedding as appropriate; maintain nutrition and hydration. Prodrome awareness and caregiver education (days of worsening bradykinesia, limb coldness, confusion); home temperature checks. Outpatient rewarming for mild hypothermia (warm room/blankets, warm oral fluids). Inpatient active rewarming for moderate–severe cases (warmed IV fluids, forced-air warming, cardiac monitoring); evaluate triggers such as infection or medication effects.	Carbidopa/levodopa for spontaneous periodic hypothermia with hypothalamic involvement. Review and withdraw offending agents (e.g., pramipexole, bromocriptine) when implicated.	Escalate for core ≤32–33 °C, arrhythmias/Osborn waves, altered mental status, recurrent episodes, or failure of home rewarming. Aim for flexible layering to handle alternating sweats/chills.

**Table 4 cells-14-01910-t004:** Management of Anhidrosis, Hypohidrosis, and Heat Intolerance.

Non-Pharmacologic Measures	Pharmacologic Measures	Clinical Notes
Cool indoor environments, breathable/light clothing, shaded spaces, and scheduling activity during cooler hours. Structured hydration plan with frequent small volumes to maintain electrolyte balance. Evaporative or portable cooling (fans, misting sprays, cooling towels/vests). Patient and caregiver education on early heat stress signs. Heat-wave preparedness: access to cooling centers and regular check-ins.	Minimize or discontinue anticholinergics that worsen sudomotor dysfunction. Review comorbid medications (e.g., nonselective β-blockers) that may impair thermoregulation. Adjust dopaminergic therapy to reduce OFF-related sweating abnormalities.	Heat intolerance is amplified by reduced or absent sweating. Dehydration greatly worsens heat sensitivity. Climate change and increased heat waves reinforce the need for proactive adaptation.

**Table 5 cells-14-01910-t005:** Management of Hyperthermia and Parkinsonism-Hyperpyrexia Syndrome.

Non-Pharmacologic Measures	Pharmacologic Measures	Clinical Notes
Active external cooling during events (ice packs to neck, axillae, or groin; fans plus mist). Evaporative or forced-air cooling as hospital protocols for moderate–severe cases. Ensure device continuity: timely DBS battery replacement, avoid prolonged deactivation, verify continuous infusion systems (Duopa/Vyalev). Maintain structured hydration and cooling during inpatient care or high-risk periods.	Rapid dopaminergic reinstatement for Parkinsonism–Hyperpyrexia Syndrome (oral, NG, transdermal, or subcutaneous routes). Supportive therapy with fluids, electrolytes, and infection management. Recognize that antipyretics and dantrolene are ineffective in PHS, prioritize dopaminergic rescue.	Suspect PHS after medication or device interruption with fever, rigidity, autonomic instability, or altered mental status. Escalate urgently for T ≥ 40 °C, rising CK, confusion, or failed outpatient cooling. Coordinate immediate communication with neurology for DBS or infusion system malfunction or shutdown during PHS events. Escalate to care team for consideration of treatment of hardware malfunction or end of life battery in parallel with colling rescue strategies.

**Table 6 cells-14-01910-t006:** Management of Medication-Related Thermoregulatory Effects.

Non-Pharmacologic Measures	Pharmacologic Measures
Patient/caregiver education on hydration, heat avoidance, early recognition of dysregulation Environmental cooling during drug-induced heat stress	Minimize/avoid anticholinergics that suppress sweating Adjust dopaminergic therapy to prevent abrupt withdrawal and lethal hyperthermia Recognize dopamine agonist–induced hypothermia (e.g., bromocriptine) and antipsychotic-induced hypothermia; discontinue offending drugs

## Data Availability

Not applicable.

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
