# Peer review of "Thermoregulatory Dysfunction in Parkinson’s Disease: Mechanisms, Implications, and Therapeutic Perspectives"

_cells, 2025, doi:10.3390/cells14231910_

Round 1
Reviewer 1 Report
Comments and Suggestions for Authors
a good effort for a review paper.
in terms of the introduction, discuss a little more about the type of nucleus involved and where do you suppose the sympathetic and parasympathetic system with sunuclean deposition
in the results, you may want to discuss multisystem atrophy and how does this differ from Parkinson’s disease and is there thermal regulation problems with this disease also.
you may also want to know if medication’s affect thermal regulations, such as amantadine.
you may also want to discuss term regulation as a dysautonomic function, for Parkinson’s disease.
when you talk about the results and conclusions you may want to discuss limitations such as in the future what study the effective medications and also order comorbidities such as a typical Parkinson’s disease diagnosis Parkinson’s disease,.
a very good effort.
Author Response
Comment 1: in terms of the introduction, discuss a little more about the type of nucleus involved and where do you suppose the sympathetic and parasympathetic system with sunuclean deposition
Response 1: Thank you for this insightful suggestion. We agree that clarifying the specific autonomic nuclei involved in α-synuclein deposition strengthens the context for understanding thermoregulatory dysfunction in Parkinson’s disease. In response, we have added a sentence to the Introduction identifying the sympathetic and parasympathetic nuclei affected—including hypothalamic preoptic regions, brainstem autonomic centers, intermediolateral spinal columns, and postganglionic efferents—and describing how degeneration in these structures contributes to impaired thermoregulation. The added sentence reads: “α-synuclein deposition affects both sympathetic and parasympathetic nuclei, including hypothalamic preoptic regions, brainstem autonomic centers, intermediolateral spinal columns, postganglionic efferent, and could contribute to the functional alteration of these centers in PD.”
Comment 2: in the results, you may want to discuss multisystem atrophy and how does this differ from Parkinson’s disease and is there thermal regulation problems with this disease also.
Response 2: Thank you for this thoughtful suggestion. We agree that multiple system atrophy (MSA) is relevant given its autonomic involvement, but it differs substantially from PD/DLB in both pathology and lesion distribution. In accordance with the reviewer’s recommendation, we removed the previously added concluding sentence and instead added a clarifying statement in the Introduction noting that although MSA shares autonomic manifestations, its predominantly preganglionic pathology and distinct α-synuclein distribution warrant separate consideration. Because literature specifically addressing thermoregulation in MSA is limited, we clarified that this review will focus primarily on PD/DLB. The added sentence reads: “However, while MSA shares some clinical features of thermoregulatory dysfunction with PD, its predominantly a preganglionic pathology and distinct α-synuclein distribution make it a separate entity; given the limited research specifically addressing thermoregulation in MSA, this review will focus primarily on PD/ Dementia with Lewy Bodies (DLB) while noting that overlapping mechanisms among these pathologies remain plausible.”
Comment 3: you may also want to know if medication’s affect thermal regulations, such as amantadine.
Response 3: Thank you for raising this point. We reviewed the available evidence regarding amantadine and thermoregulation. Although classic animal studies demonstrate that amantadine produces a marked, centrally mediated hypothermia in rodents and that this effect is not dopamine-receptor dependent, no corresponding human data are establishing a thermoregulatory effect in Parkinson’s disease. Because our review focuses on clinically supported mechanisms in humans, we added a brief clarifying sentence to Section 6.5, noting that the thermoregulatory findings for amantadine are limited to animal models and that its relevance to PD patients remains uncertain. No additional discussion was added, as current evidence does not support a defined clinical effect. The added sentence reads: “Although early animal studies show that amantadine can induce centrally mediated, dopamine-independent hypothermia in rodents, comparable effects have not been demonstrated in humans, and its relevance to thermoregulation in PD remains unclear (107).”
Comment 4: you may also want to discuss term regulation as a dysautonomic function, for Parkinson’s disease.
Response 4: Thank you for this valuable comment. To improve conceptual clarity, we added a sentence in the Introduction situating thermoregulatory dysfunction within the broader framework of dysautonomia in PD. This addition appears alongside historical descriptions of autonomic dysfunction. The added sentence reads: “Additional autonomic cardiovascular and sexual dysfunction are also identified in PD.”
Comment 5: when you talk about the results and conclusions you may want to discuss limitations such as in the future what study the effective medications and also order comorbidities such as a typical Parkinson’s disease diagnosis Parkinson’s disease.
Response 5: Thank you for this important comment. We agree that acknowledging limitations and highlighting future directions related to medication effects and atypical parkinsonian comorbidities would strengthen this section. To address this, we have added a sentence to the “Gaps and Areas for Future Research” section the need for studies that evaluate how medications impact thermoregulatory function and how these issues may differ across atypical Parkinsonian disorders with distinct autonomic profiles. This addition clarifies the scope of unmet research needs without expanding beyond the evidence summarized in the manuscript. The added sentence reads: “Future work should also address major knowledge gaps in how medications influence thermoregulatory function and evaluate these responses across atypical parkinsonian disorders, where differing patterns of autonomic involvement may alter therapeutic needs and outcomes.”
Reviewer 2 Report
Comments and Suggestions for Authors
This review article comprehensively explains thermoregulatory dysfunction in Parkinson’s disease (PD). Although thermoregulatory disturbances are usually observed in patients with advanced-stage PD, their detailed pathophysiology is not yet fully understood, even among movement disorder specialists. This review may therefore be particularly helpful for clinicians in this field.
The following are my comments to the authors.
- In the section on normal thermoregulatory function, the physiological role of brown adipose tissue (BAT)—including its signaling cascades—should be described in greater detail, as BAT plays a key role in thermogenesis.
- Hyperhidrosis is usually difficult to treat in patients with PD. In particular, generalized hyperhidrosis that is not necessarily related to troublesome dyskinesia or elevated ambient temperature should be discussed, together with plausible underlying pathomechanisms.
- The authors state that the use of anticholinergics may raise core temperature and blunt sweating. However, currently used anticholinergics such as solifenacin, darifenacin, imidafenacin, and tolterodine are more selective for the bladder than for other organs, compared with conventionally used anticholinergics. Please add a discussion of how these newer agents may affect thermoregulation in PD.
- In the “DIAGNOSTICS AND TEST” section, figures illustrating each test would likely be helpful for readers.
- Would it be useful to assess postganglionic sudomotor function using skin biopsy specimens?
Author Response
Comment 1: In the section on normal thermoregulatory function, the physiological role of brown adipose tissue (BAT)—including its signaling cascades—should be described in greater detail, as BAT plays a key role in thermogenesis.
Response 1: Thank you for your insight. Following your suggestion, we expanded the section regarding BAT in the “Normal Thermoregulatory Function” section to include greater detail on sympathetic regulation, β3-adrenergic control, UCP1-dependent thermogenesis, and BAT’s contribution to non-shivering heat production.
Comment 2: Hyperhidrosis is usually difficult to treat in patients with PD. In particular, generalized hyperhidrosis
that is not necessarily related to troublesome dyskinesia or elevated ambient temperature should be discussed, together with plausible underlying pathomechanisms.
Response 2: Thank you for this helpful comment. We agree that distinguishing spontaneous generalized hyperhidrosis from forms triggered by dyskinesia or environmental heat strengthens the clinical clarity of this section. To address this, we have added a sentence to the Hyperhidrosis subsection noting that generalized hyperhidrosis in PD can occur independently of motor fluctuations or external temperature stressors, likely due to intrinsic autonomic dysregulation. This addition aligns with the manuscript’s emphasis on sympathetic instability as a driver of sweating abnormalities in PD. The added sentence reads: “Generalized hyperhidrosis may also emerge spontaneously in PD, independent of dyskinesias or heat exposure, likely reflecting underlying autonomic dysregulation that disrupts normal sympathetic control of sweating.”
Comment 3: The authors state that the use of anticholinergics may raise core temperature and blunt sweating. However, currently used anticholinergics such as solifenacin, darifenacin, imidafenacin, and tolterodine are more selective for the bladder than for other organs, compared with conventionally used anticholinergics. Please add a discussion of how these newer agents may affect thermoregulation in PD.
Response 3: Thank you for this insightful comment. We reviewed the pharmacologic data comparing older nonselective anticholinergics with newer bladder-selective agents. As summarized in the referenced therapeutic review, medications such as solifenacin, darifenacin, imidafenacin, and tolterodine demonstrate markedly greater uroselectivity, reduced off-target muscarinic binding, and minimal central penetration. These properties suggest a substantially lower likelihood of impairing sweating or increasing core temperature compared with conventional anticholinergics. To address this important distinction, we have added a sentence to Section 4.5 (Medication-Related Thermoregulatory Effects) noting that these newer agents are expected to have less clinically significant impact on thermoregulation in PD. The added sentence reads: “Newer anticholinergics, such as solifenacin, darifenacin, and imidafenacin, are more selective M3 acetylcholine receptor antagonists and thus exhibit higher uroselectivity and exert less disruption of sweating and core temperature control than older, nonselective agents. (84)”
Comment 4: In the “DIAGNOSTICS AND TEST” section, figures illustrating each test would likely be helpful for readers.
Response 4: Thank you for this helpful suggestion. We agree that figures illustrating each diagnostic test could aid reader comprehension. We carefully considered adding such illustrations; however, after review, we determined that incorporating figures for every test in the “Diagnostics and Tests” section would significantly expand the length of the manuscript and would largely duplicate information already condensed and organized in Table 1. Table 1 currently summarizes test principles, clinical applications, and interpretive value in a format that avoids redundancy and maintains focus. Adding multiple figures would reduce conciseness without providing substantial new information. For these reasons, we ultimately elected not to add additional illustrations. No changes were made in the manuscript in response to this comment.
Comment 5: Would it be useful to assess postganglionic sudomotor function using skin biopsy specimens?
Response 5: Thank you for raising this point. Although skin biopsy–based assessment of postganglionic sudomotor fibers is a promising avenue, its current clinical and research utility remains limited. At present, even if a biopsy were performed, there are not yet standardized or widely validated methods to convert those findings into reliably interpretable or clinically actionable data. Because of these limitations, referencing this technique into the manuscript would introduce a level of methodological speculation that does not align with the practical, currently implementable diagnostics emphasized in our review.
Reviewer 3 Report
Comments and Suggestions for Authors
Author discussed and described comprehensive views and literature about association and impact of thermoregulation in Parkinson's disease. It is interesting topic to understand and looks towards the Parkinson's disease. Authors did well to summarize all aspects. However, I suggest that Author may make it more easy for the readers by introducing more illustration about the diagnostic tools or tests if possible.
Author Response
Comment 1: If it is possible, then Authors may demonstrate diagnostics and test methodology by describing through illustration. It is easier to understand the diagnostic tools or methods.
Response 1: Thank you for pointing this out. We appreciate the reviewer’s suggestion to include diagnostic illustrations to demonstrate the testing methodology. We carefully considered adding such illustrations. However, after review, we determined that incorporating additional schematic figures would not be feasible within the current manuscript format and would largely duplicate information that is already presented in Table 1, which summarizes diagnostic tools, pathways, and interpretation in detail (p. 15 of the manuscript). Therefore, we have opted not to add new illustrations. At present, we believe the table offers the most concise and accessible presentation of diagnostic methodology for readers.
Round 2
Reviewer 2 Report
Comments and Suggestions for Authors
Thank you for your revision. I think that the manuscript was revised correctly.